# Root Rot of *Cinnamomum camphora* (Linn) Presl Caused by *Phytopythium vexans* in China

**DOI:** 10.3390/plants12051072

**Published:** 2023-02-28

**Authors:** Yatong Xiao, Min Li, Fengmao Chen

**Affiliations:** Co-Innovation Center for Sustainable Forestry in Southern China, College of Forestry, Nanjing Forestry University, Nanjing 210037, China

**Keywords:** *Cinnamomum camphora*, *Phytopythium vexans*, root rot, identification, fungicides

## Abstract

As a famous street tree, camphor (*Cinnamomum camphora*) is widely planted worldwide. However, in recent years, camphor with root rot was observed in Anhui Province, China. Based on morphological characterization, thirty virulent isolates were identified as *Phytopythium* species. Phylogenetic analysis of combined *ITS*, *LSU rDNA*, *β-tubulin*, *coxI,* and *coxII* sequences assigned the isolates to *Phytopythium vexans*. Koch’s postulates were fulfilled in the greenhouse, and the pathogenicity of *P. vexans* was determined by root inoculation tests on 2-year-old camphor seedlings; the symptoms of indoor inoculation were consistent with those in the field. *P. vexans* can grow at 15–30 °C, with an optimal growth temperature of 25–30 °C. The results of fungicide sensitivity experiments indicated that *P. vexans* was the most sensitive to metalaxyl hymexazol, which may be a useful idea for the future prevention and control management of *P.vexans*. This study provided the first step for further research on *P. vexans* as a pathogen of camphor, and provided a theoretical basis for future control strategies.

## 1. Introduction

The camphor tree [*Cinnamomum camphora* (Linn.) Presl] is an important species of subtropical evergreen broad-leaved tree [1] that is widely distributed in Asia [2]. The tree is native to China and is often regarded as both an ornamental tree and a street tree because of its tallness and beauty [3]. Camphor is economically valuable in many ways. As one of the oldest traditional herbal medicines, it has antibacterial, antioxidant and other biological functions [4,5,6]. Camphor is usually used as a raw material for producing camphor pills [4], wood [4], essential oils [7], and perfumes [8]. Medicinal products made from camphor trees can be used to treat inflammation-related diseases such as sprains, bronchitis and rheumatism, and camphor leaves can also be made into camphor balls with different chemical components [7,9]. However, in the nursery land of the city of Xuancheng, Anhui Province, China, a large number of camphor trees were found to have root rot, which resulted in the death of camphor trees across a large area. This phenomenon has placed great pressure on the local environment and caused economic losses.

Oomycetes are a unique group of fungus-like eukaryotic microbes, many of whose genera are potential threats to the healthy growth of plants and animals [10]. As a group, they are best known as plant pathogens. Two of the more notorious genera of oomycetes are *Pythium* and *Phytophthora*, which cause economic losses to crops and plants and continue to have a devastating impact on natural ecosystems [11]. It is important to note that *Phytophthora cinnamomum* is an important root pathogen with a wide host range of more than 900 species of host plants worldwide [12]. *Pythium* also contains destructive species which cause root rot, seedling wilt, and stem rot in agronomic and vegetable crops [13]. Economic losses caused by oomycetes are estimated to exceed billions of dollars worldwide. In cases in which an oomycete becomes established in nursery soil, it can influence the roots of the plant and spread through various routes [14].

In the past, the use of protective fungicides has not been sufficient to control the development of soilborne oomycetes [15]. At the end of the last century, metalaxyl, dexon and etridizole began to be applied to control *Pythium* in the field; these fungicides can sterilize seeds and also be sprayed directly in the soil to play a preventive role [16]. Currently, there are several different chemical groups that control oomycetes, such as phenylamides (PAs), dithiocarbamates (e.g., mancozeb), and chlorothalonil [17]. Oxathiapiprolin and fluopicolide have recently been considered to be resistant to some genera of oomycetes, such as *Pythium* and *Phytophthora*; they have been shown to have a significant inhibitory effect on the release and activity of zoospores, mycelial growth and the sporulation ability of oomycetes [18,19]. 

*Phytopythium* is a recently established genus between *Phytophthora* and *Pythium* in terms of age, with similar characteristics. The sporangia of *Phytopythium* are more similar to those of *Phytophthora*, and the mode of discharge of zoospores, zoospores being created in the vesicle, is similar to that of *Pythium* [20,21,22]. The classification of taxa in the genus *Pythium* was revised on the basis of morphological differences in sporangia and the phylogenetic analysis of the *LSU rDNA* and *coxI* mDNA regions, from which four new genera were established: *Ovatisporangium*, *Globisporangium*, *Elongisporangium*, and *Pilasporangium* [23]. *Ovatisporangium* is treated with the same name as *Phytopythium* [24]. *Phytopythium vexans* has caused crown and root rot in *Prunus serrulata* Lindl. in Tennessee [25] and widespread leaf blight and root rot of *Manihot esculenta* in Brazil [26].

This study presents the features of oomycetes associated with root rot of camphor trees in Anhui Province, China, which caused serious damage and placed pressure on the local ecological environment. The objectives of this study were as follows. First, we aimed to identify oomycete pathogens that cause root rot in camphor trees by Koch’s postulates. Second, molecular biology and morphological identification were used to identify the pathogen. Third, fungicides with obvious inhibitory effects on the mycelial growth of the pathogen were screened by a media plate phenotyping experiment.

## 2. Results

### 2.1. Field Survey, Oomycete Isolation and Purification

Early in the second half of 2021, sixty camphor trees began to die, with symptoms first appearing in September and developing more rapidly in autumn. By the end of 2021, the incidence of disease in the field was about 88.3% (265 symptomatic trees), and about 27% of the 300 camphor trees surveyed died (81 dead trees). The pathogen can infest seedlings at different times of the year and cause tree death within a relatively short period of time, placing considerable strain on the ecological environment. Under normal growth, camphor is an evergreen tree year-round; however, after infection with the pathogen, the leaves gradually fade to green and yellow (Figure 1A). At a later stage, the leaves crumpled and easily fell off until the whole tree died (Figure 1B,C). The branches and stems become black and necrotic, and eventually, the whole tree dies. The xylem appears brown in longitudinal sections, and the root epidermis rots and falls off (Figure 1D,E).

In this study, one oomycete was consistently isolated from all tissue masses in the 30 root samples investigated.

### 2.2. Molecular Identification and Phylogenetic Analysis

All sequences obtained in this study were compared with *Phytopythium* isolates available in GenBank, and the homology of all isolates was verified by BLAST calculation of nucleotide identity. All sequences of 30 isolates were deposited in GenBank, and the accession numbers for the sequences of the 31 reference *Phytopythium* sp. isolates in the GenBank database are presented in Table 3. To produce the phylogenetic tree, we used a total of 294 sequences from 62 isolates. Phylogenetic analysis was based on sequences from 5 genomic regions: *ITS*, *LSU rDNA*, *β-tubulin*, *coxI* and *coxII*. For these loci, fragments of 951 bp, 683 bp, 715 bp, 670 bp and 534 bp were obtained by PCR amplification and bidirectional sequencing.

Two separate tree-building methods were used to sequence and analyze individual gene sequences before building a phylogenetic tree with five sets of gene chains. By comparison, no apparent conflicting contradictions were found in the individual gene phylogenies, so the *ITS*, *coxI*, *coxII*, *LSU rDNA* and *β-tubulin* datasets could be combined and concatenated. Cluster analysis was performed using sequences downloaded from the NCBI database. The concatenated matrix contained 3553 bp nucleotides. The tree topology obtained by Bayesian analysis and ML analysis was basically the same, which indicated that the evolutionary relationship of the isolates was statistically supported (Figure 2). Clustering of isolates obtained from camphor samples with the reference strain *P. vexans* was statistically supported by phylogenetic analysis, with 99% bootstrap proportions and a 1 BPP, resulting in a separate clade within the *Phytopythium* fraction (Figure 2). The isolates obtained from camphor samples were closely related to *P. vexans* and clustered with *P. vexans* isolate 2D111, with statistical support of 100% BP and 1 BPP. Phylogenetic analysis showed that all the isolates obtained in this study were highly similar to the previously reported isolates of *P. vexans*. Therefore, the pathogen was confirmed to be *P. vexans*.

### 2.3. Morphological Identification and Biological Characteristics

The colonies of four representative isolates grew rapidly and the colonies covered the surface of 90 mm V8-agar medium in 3 days at 25 °C. Morphological features were recorded based on visual observation. The isolates showed white colonies on CMA medium with no obvious radiolucent pattern (Figure 3A,F), and grew better on PDA medium, with colonies showing a fluffier and richer mycelium with a radial morphology (Figure 3B,G). Those cultured on V8–agar medium had a velvety blooming colony morphology with no visible sporangium formation on the surface (Figure 3C,H). However, the abundance of aerial hyphae varied among the types of medium. The hyphae on the PDA medium presented a petal-like and radial shape, and the isolates had the densest and most abundant mycelia on the V8–agar and PDA media. After 3 days of culture, the colonies on the PDA medium radiated in the form of rose petals, and the hyphae gradually became dense and velvet-like from the center to the edge; the hyphae were irregularly dispersed on the PCA medium (Figure 3D,I). The growth of the colonies on GPYA medium was weak, but the hyphae were also abundant (Figure 3E,J).

The isolates have distinct papillae (or protrusions) and proliferating free sporangia inside. After 3 days of water incubation, these traits were observed under a microscope. The shape of the sporangia was round to ovoid, with a smooth surface, and a rich matrix could be seen inside the sporangia (Figure 4B,C). It was also possible to observe empty sporangia (Figure 4A). Some of the oospores were surrounded by oogonia or showed short protuberances (Figure 4D–F). The average size of the sporangia was 13.19 × 12.41 μm, and the size of the sporangia ranged from 12.62–16.36 × 8.57–16.45 μm (n = 100). Based on colony morphology, color and sporangia, the isolates were identified as *P. vexans*.

The isolates grew in the range of 15–30 °C, and 25–30 °C was the optimal temperature for growth (Figure 5). The colony hyphae were most dense and fluffy when the temperature range was 25–30 °C, and the colony did not grow when the temperature reached 35 °C (Figure 6).

### 2.4. Pathogenicity Tests

*Phytopythium vexans* (representive isolates ZS01, ZS02, ZS03 and ZS04) caused symptoms of camphor wilt at 14 days after inoculation. Nevertheless, no symptoms occurred in the control group. Fourteen days after inoculation, the plants in the control group grew vigorously, with emerald-green leaves spreading outward (Figure 7A). The inoculated plant leaves began to droop with symptoms of wilting (Figure 7B). Then, 21 days after inoculation, the symptoms gradually worsened, the degree of blight increased each day, and death occurred on day 38 (Figure 7C–E). Compared with that of the control seedlings, the growth vigor of the inoculated camphor seedlings became weaker, while the uninoculated camphor seedlings grew healthy, and the root vigor of the inoculated seedlings was weakened and degraded. No symptoms were found on uninoculated plants (Figure 7F), and the root tissue was robust (Figure 7F). The longitudinal section of the stem of the camphor inoculated with isolates was dark and dull in color (Figure 7H), and the xylem stripes in the longitudinal section of the stem were clear and healthy **(**Figure 7G). The leaves of camphor seedlings inoculated with isolates turned yellow and withered, and the roots decayed and were fibrous and easily detached. The longitudinal section of the roots was rough and accompanied by darkening symptoms (Figure 7J), and the surface of the longitudinal section of the root of the control group was clear and bright (Figure 7I). The reisolations were recovered from symptomatic root tissue, and the cultures were similar in character and morphology to the isolates. DNA sequences extracted from reisolation obtained from inoculated roots matched the DNA sequences of the isolates used for inoculation, and the symptoms on artificially inoculated seedlings were similar to those in the field, thus satisfying Koch’s postulates.

### 2.5. Susceptibility of Phytopythium Isolates to Fungicides

The four representative strains showed similar biological responses to the five fungicides (Figure 8). All five fungicides showed significant growth inhibition on the representative isolates on V8-agar media; metalaxyl hymexazol had lower EC_50_ on mycelial growth than the other four fungicides, and difenoconazole had the highest EC_50_ on mycelial growth of the representative isolates, and the weakest inhibition effect (Table 1). These results indicated that metalaxyl hymexazol was the most effective fungicide against *Phytopythium* sp. in this study.

## 3. Discussion

The pathogen that causes root rot can not only infect camphor, but can also infect other potential host plants in the nursery. It is very necessary to identify the pathogen that causes root rot of camphor in a timely manner. Based on morphological identification and molecular and phylogenetic analysis, *Phytopythium vexans* was identified as the causal agent of root rot on camphor in China. According to previous studies, there have been no reports about *P. vexans* causing root rot on camphor in the world.

*Pythium* is usually classified based on morphological characteristics, such as the shape and size of sporangia and oogonia, and the number and position of antheridia [27,28]. However, these characteristics are very similar among the members of *Pythium*, and it is difficult to classify and identify *Pythium* by morphological characteristics alone. Based on previous studies, to date, phylogenetic analysis of *Pythium* is mainly based on the rDNA large subunit (*LSU rDNA*), *ITS*, *β-tubulin*, and cytochrome oxidase II (*coxII*) gene sequences [29,30,31,32,33,34]. In this study, phylogenetic analysis of a combination of *ITS*, *LSU rDNA*, *coxI*, *coxII*, and *β-tubulin* sequence data indicated that isolates from all collected samples were single species. The phylogenetic tree showed that the *P. vexans* obtained from the roots of camphor in this study were similar to those reported in other studies (Figure 2).

*Phytopythium vexans* (formerly known as *Pythium vexans*) is distributed worldwide and can cause root rot, stem rot, crown rot and leaf ulcers in many woody ornamentals [35]. According to previous academic research in South Africa, the pathogen mainly causes diseases in woody plants and economic crops, such as apples [36] and kiwifruit [37]. The incidence of brown root rot of ramie caused by *P. vexans* has caused over 40% of production losses in China [38]. In addition to seriously affecting cash crops such as fruit, it can also harm woody ornamentals such as ginkgo and red maples [35]; it has also been isolated from infected *Anthurium andraeanum* in Korea [39]. To summarize, the ecological loss caused by *P. vexans* worldwide is huge, so the ecological threat of *P. vexans* to the environment cannot be underestimated.

Temperature is environmental factor that is generally considered to be the main factor affecting the prevalence of plant diseases [40]. In this study, the optimum growth temperature of the representative isolates ranged from 25 °C to 30 °C; this is consistent with the results of previous studies on the optimum growth temperature for *P. vexans* [41,42]. Therefore, timely intervention of pathogens should be conducted before the appropriate growth temperature is reached. There were abundant water sources around the nursery site, and the environment was relatively humid, which may explain the prevalence of pathogens in the nursery site. Such pathogens will proliferate in a high-humidity environments, produce sporangia, release highly infectious zoospores when the conditions are suitable, and begin to infect suitable host plants. This finding indicates that future field management measures should include appropriate irrigation methods according to the growth characteristics and needs of crops. At the same time, the health of water sources is also one of the important factors preventing infection by this pathogen. It is well known that water is the main substrate for *P. vexans*, and a drop of water can spread zoospores within a radius of one meter because the pathogen is prone to encounter suitable environmental conditions in water, thereby releasing infectious zoospores [43]. From the perspective of controlling the development of disease, determining the biological characteristics of pathogens is very important for the prevention and control of plant disease epidemics, which can provide a scientific basis for disease prevention and control. The temperature experiment in this study proved that the time of application of fungicide should be determined before the optimal growth temperature of the pathogen is reached.

*Phytopythium vexans* is a soil-borne pathogen that has a wide range of transmission pathways in nurseries and spreads rapidly. The presence of the pathogen should be detected as soon as possible to help prevent the occurrence of the disease [44]. Soil-borne diseases can affect the health and appearance of a plant and even its economic value, so timely intervention is essential. There are various management measures for soil-borne diseases, such as improving the environmental conditions of nursery land, introducing relevant laws and regulations, breeding highly resistant varieties, and implementing biological and chemical control [45]. When the pathogen begins to infect plants in the soil of the field, it is very necessary to inhibit the growth of mycelial and sporangium production in a timely manner. In 2020, soybeans grown in Huang-Huai area of China have been infected by *Pythium* and *Phytopythium*. In the control of the dominant strains of the isolates, the researchers selected the fungicides containing metalaxyl, and the EC_50_ values obtained were all less than 1 μg/mL. Based on the experiments of this study, metalaxyl hymexazol had the best growth inhibition effect on *P. vexans*, and the EC_50_ value was less than 0.01 μg/mL. In 2022, soybean growing areas in the United States were also affected by some species of *Phytophthora*, *Pythium*, and *Phytopythium*, and researchers used a fungicide mixed with oxathiapiprolin and metalaxyl to control oomycetes more effectively than oxathiapiprolin alone. [46,47]. This indicates that in future field control, appropriate use of fungicides containing metalaxyl can effectively prevent the spread of oomycete pathogens. The next step is to quickly determine the concentration of effective fungicides to maximize the protection of plants from these pathogens, and then test it on the host plant to obtain the optimal application concentration.

As an important roadside and ornamental tree, camphor is widely planted in China. The isolation and identification of the pathogenic species described herein provide new ideas and references for the cultivation and management of camphor trees in China. Regarding camphor root rot, there are many issues that need to be further studied and solved. *P. vexans* may pose a great threat to the environment in the future, as the pathogen can infect a wider range of hosts. Testing of the soil in and around the sites wherein camphor root rot occurs and rational use of fungicides and fertilizers are both recommended in order to actively maintain appropriate environmental conditions and prevent the occurrence of the disease.

## 4. Materials and Methods

### 4.1. Field Survey, Oomycete Isolation and Purification

In 2021, planted camphor plots in the city of Xuancheng (118°75′, 30°94′), Anhui Province, China, were surveyed for the occurrence of root rot. The 9–10 year old camphor trees in the planting area (300 trees) were investigated, and the external symptoms of tree diseases were recorded. Disease incidence was calculated by counting the number of symptomatic trees, asymptomatic trees, and dead trees. The local area is more loosely managed, with a high level of human activity and low-lying terrain that is prone to waterlogging, which is not conducive to the growth of camphor trees. Thirty symptomatic tissues obtained from the roots of diseased plants were washed first in tap water and then in sterile distilled water and cut into small pieces of 0.5–1 cm^2^. These small pieces were soaked in 75% ethanol for 45 s for surface disinfection and then drained with sterile filter paper. Pieces were transferred to V8 (vegetable juice)-PARP-agar medium [48] in 90 mm-diameter Petri dishes and incubated at 28 °C in the dark for 4 days. The PARP contained pimaricin (20 mg/L), ampicillin (125 mg/L), rifampin (10 mg/L), and pentachloronitrobenzene (20 mg/L) [49]. Pure cultures were obtained by transferring the mycelial of the margins of the colonies to V8-agar medium [48].

### 4.2. Molecular Identification and Phylogenetic Tree of Phytopythium *sp.*

The genomic DNA of isolates’ strains was obtained from mycelial colonies using the cetyltrimethylammonium bromide method (CTAB) [50]. In short, a small amount of mycelium was cut into a 2 mL sterile centrifuge tube, the bottom of the tube was filled with 500 mL 2% CTAB and 500 mL chloroform, and the tube was placed in a shaker at 200 r/min at 25 °C for 1.5 h and centrifuged at 13,000 r/min for 15 min after removal. After centrifugation, 300 μL of the supernatant was transferred to another 1.5 mL sterile centrifuge tube containing 600 μL absolute ethanol, and then centrifuged at 13,000 r/min for 5 min. The supernatant was discarded, and 1 mL 75% ethanol was added for elution twice, followed by centrifugation at 13,000 r/min for 5 min. Then, the solution was placed in an oven (at 65 °C) to wait for the ethanol to evaporate, and 30 μL sterile deionized water was added for precipitation to obtain a crude DNA suspension.

The primer pairs used for sequence amplification of the rDNA internal transcribed spacer (ITS), large subunit (LSU rDNA), β-tubulin and cytochrome c oxidase I and II (coxI and coxII) genes are listed in Table 2. Each 50 μL reaction mixture contained 25 μL of Green PCR Master Mix, 17 μL of sterile deionized water, 4 μL of DNA, and 2 μL each of the upstream and downstream primers. The PCR products were electrophoresed (150 V for 25 min) on 2% agarose gels and sequenced at the Shanghai Sangon Biological Technology Company. All sequences obtained in this study were uploaded in GenBank (Table 3). 

The sequences derived from the isolates in this study and the sequences related to the *Phytopythium* sp. in Genbank were used to construct the phylogenetic tree; Phytophthora nicotianae was used as the outgroup. (Table 3). BioEdit version 7.0.9.0 software was used to align the nucleotide sequences, and the missing bases in these sequences were manually corrected [51]. Phylogenetic trees of combined genes were constructed using two independent optimality search criteria: Bayesian inference (BI) and maximum likelihood (ML) criteria. The ML analysis was performed using IQ-TREE [52], and the GTR+G+I model was chosen to estimate branch stability by 1000 bootstrap replicates. The BI analysis was performed using PhyloSuite version 1.2.2. [53] under a partition model, and FigTREE version 1.4.4 was used to view the phylogenetic trees.

### 4.3. Morphological Observations and Biological Characteristics

Four isolates (ZS01, ZS02, ZS03 and ZS04 based on preliminary phylogenetic analyses) were selected for morphological observation and biological characterization.

Isolates were cultured on potato dextrose agar medium (PDA), corn meal agar medium (CMA), potato carrot agar (PCA) medium, V8–agar medium and peptone yeast glucose agar medium (GPYA) in the dark at 25 °C for 3 days, and the morphology and color of the colonies were recorded. The identification of oomycete pathogens was initially based on the observation of morphological characteristics, as well as the characteristics of sporangia. To induce the production of sporangia, five plugs (2 mm × 2 mm) of isolates were transferred to 10% V8 liquid medium and cultivated for 3 days until the mycelial plugs became mycelial mats. Then, the V8 liquid was replaced with sterile water. To stimulate sporangial production, five drops of soil extract solution were added to each medium [48]. This operation was repeated for approximately 3 days, and the hyphae produced many sporangia. The oomycete structures of four isolates (ZS01, ZS02, ZS03 and ZS04) were examined and recorded. To observe more subtle features, the sterile water carrying the sporangia hyphae was fixed on a glass slide, and a Zeiss Axio imager A2m microscope was used to observe and measure the size of the sporangia at a magnification of 40×. Over 50 sporangia were randomly observed per isolate, and the experiment was repeated twice.

To determine the optimal growth temperature of the isolates, mycelial plugs (6 mm diameter) were placed on fresh V8–agar medium (90 mm diameter) and incubated from 15 to 35 °C at 5 °C intervals. Experiments were carried out at five temperatures, with five replicates per isolate. The colony growth diameter was measured and recorded daily. These experiments were conducted twice. 

### 4.4. Pathogenicity Test

Four isolates selected for morphological and biological identification were used in pathogenicity tests. Pathogenicity testing of isolates was carried out on healthy 2-year-old camphor trees. The isolates were incubated on V8–agar medium for 3 days, and some of the mycelial plugs were transferred to 10% V8 liquid for 3–5 days until abundant mycelial mats were produced. To stimulate sporangia production, the V8 liquid was decanted, and then appropriate amounts of sterile water and soil extract solution were added. The operation was repeated three times at 24 h intervals until sporangia were observed under the microscope. The sporangia suspension was inoculated onto the roots of the camphor, and plants treated with sterile V8 liquid disks were used as controls. The inoculated plants were placed in high humidity and a constant temperature of 28 °C for observation. There were five replicates per treatment and control group. The experiments were conducted three times simultaneously.

To fulfil Koch’s postulates, symptomatic tissue sections were excised from the root margins and transferred to a V8-PARP–agar medium for reisolation, and the isolation was confirmed by morphological identification and molecular identification. The primers used for molecular identification were *ITS*, *LSU rDNA*, *β-tubulin*, *coxI* and *coxII* (Table 2).

### 4.5. Susceptibility of Phytopythium Isolates to Fungicides

Metalaxyl mancozeb [68% active ingredient (a.i); Syngenta Investment Co., Ltd., Shanghai, China], metalaxyl hymexazol (30% a.i; Zhongnonglihua Agricultural Chemicals Co., Ltd., Tianjin, China), fluopicolide (68.75% a.i; Bayer Cropscience Co., Ltd., Beijing, China), difenoconazole (10% a.i; Syngenta Crop Protection Co., Ltd., Nantong, China), and oxalite mancozeb (64% a.i; Syngenta Crop Protection Co., Ltd., Suzhou, China) were used in this study. Based on the previous experiment, four representative isolates were selected. The susceptibility of each isolate to fungicides was determined on fresh V8-agar media with fungicide added. Several 5mm-diameter mycelium plugs were removed from the edge of the actively growing culture colony and placed on V8-agar media with or without fungicide (as a control). For metalaxyl mancozeb and fluopicolide, final concentrations of 0.25, 0.5, 1, 2, 4, 8, and 16 μg/mL were added to the amended media; for metalaxyl hymexazol, final concentrations of 0.2, 0.5, 1, 2, 10, 20, and 50 μg/mL were added to the amended media; for difenoconazole, final concentrations of 1, 5, 10, 20, 50, 100, and 200 μg/mL were added to the amended media; and for oxalite mancozeb, final concentrations of 1, 2, 4, 8, 16, 32, and 64 μg/mL were added to the amended media. Five replications were carried out for each treatment, and the whole experiment was repeated three times, with mean colony diameters measured 3 days after inoculation. The formula to calculate the inhibition rate is [1 − (colony diameter at fungicide concentration/colony diameter of the control)] × 100%. EC_50_ values (the concentration inhibiting growth for 50%) were estimated by regression to the log_10_ probability conversion of percentage inhibition to fungicide concentration.

## Figures and Tables

**Figure 1 plants-12-01072-f001:**
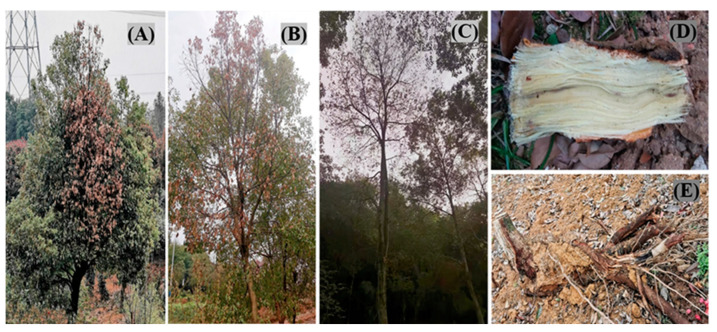
Symptoms of dieback and root rot disease on camphor (*Cinnamomum camphora* (Linn) Presl). (**A**–**C**). Diseased tree and dead plants in the nursery land of Xuancheng city, Anhui, in 2021. (**D**). Xylem browning in diseased trees. (**E**). Root rot of diseased trees.

**Figure 2 plants-12-01072-f002:**
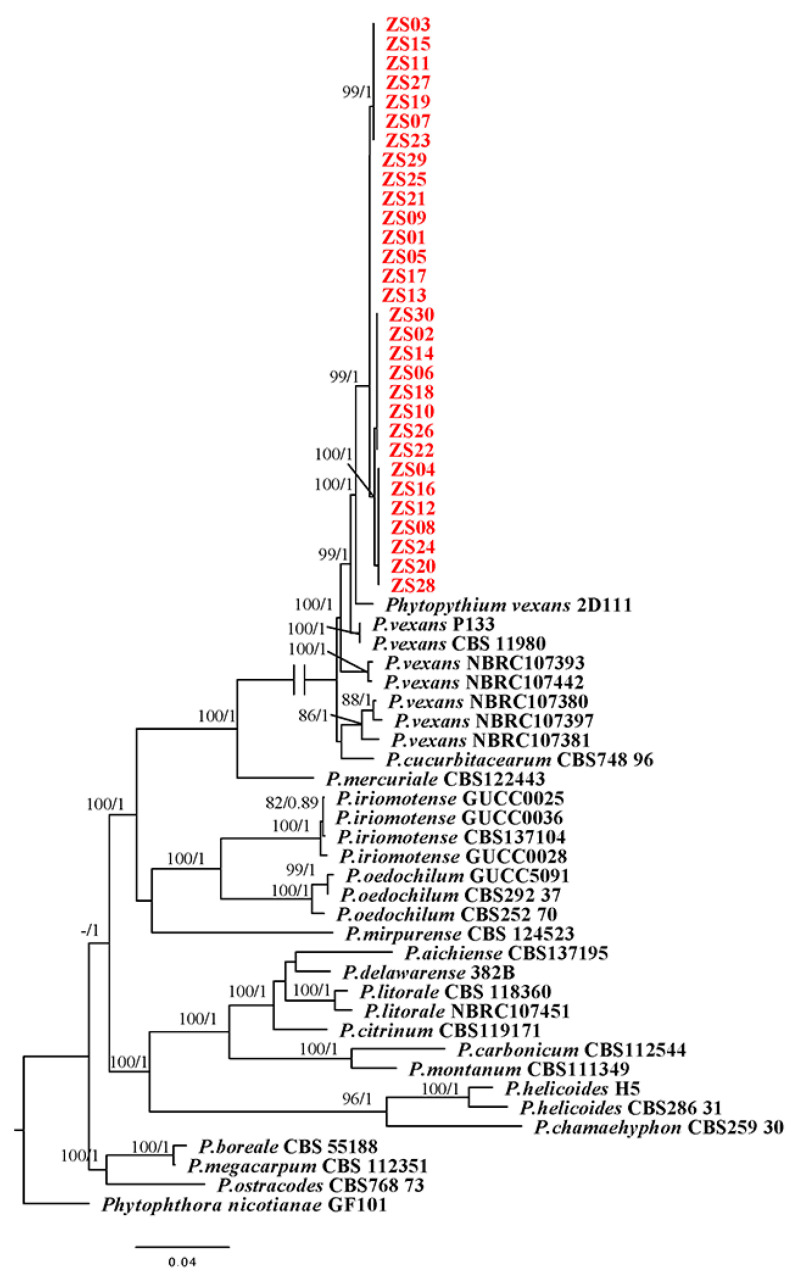
Phylogenetic tree obtained from Bayesian analysis of the combined *coxI*, *coxII*, *ITS*, *LSU rDNA* and *β-tubulin* sequence alignment of all accepted species of *Phytopythium*. The bootstrap support values and Bayesian posterior probabilities are given at the nodes (MLBS/BPP). The red font indicates the isolates obtained in this study. ZS01-ZS30 were the isolates in this experiment. Bootstrap values below 70% and posterior probabilities below 0.70 are not shown.

**Figure 3 plants-12-01072-f003:**
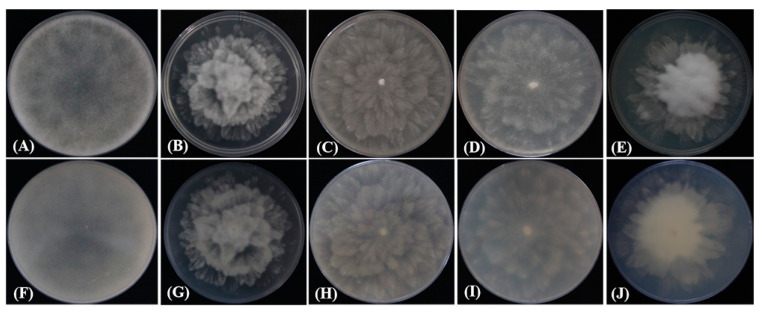
Colony formation of strain ZS01 isolated from camphor cultured after 3 days on different media at 25 °C. (**A**) Top and (**F**) reverse views of colony morphology on CMA medium. (**B**) Top and (**G**) reverse views of colony morphology on PDA medium. (**C**) Top and (**H**) reverse views of colony morphology on V8–agar medium. (**D**) Top and (**I**) reverse views of colony morphology on PCA medium. (**E**) Top and (**J**) reverse views of colony morphology on GPYA medium.

**Figure 4 plants-12-01072-f004:**
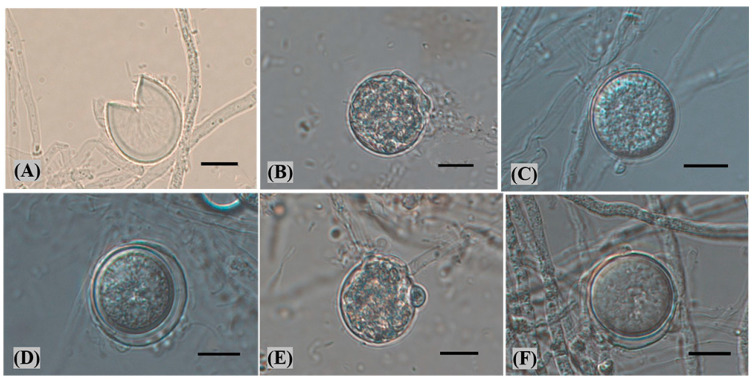
(**A**) Empty sporangia. (**B**,**C**) Sporangia. (**D**,**F**) Sexual structures showing an oogonia and oospore. (**E**) Sporangia showing short protuberances. Bars = 10 μm.

**Figure 5 plants-12-01072-f005:**
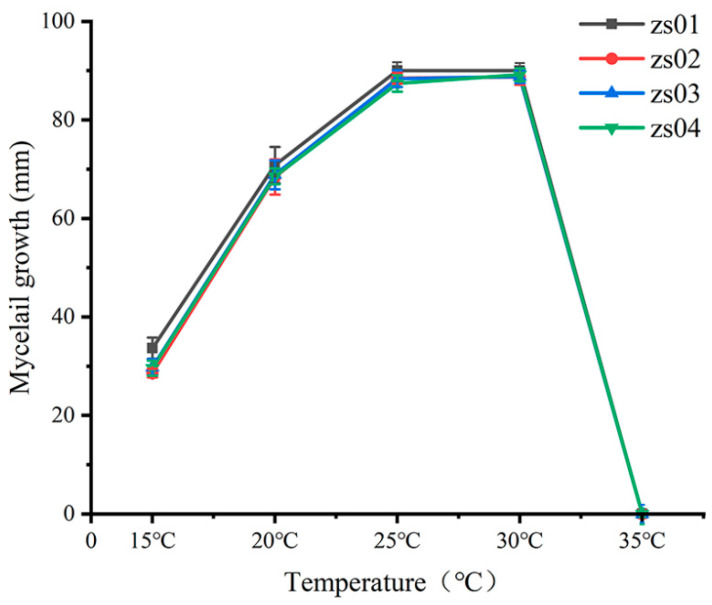
Effect of temperature on colony diameter growth of isolates ZS01–04 after 3 days of culture on V8-agar media.

**Figure 6 plants-12-01072-f006:**
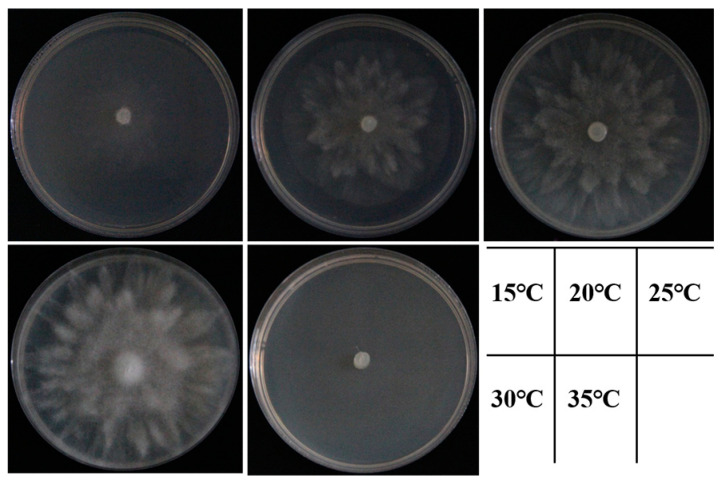
Colony morphology of the isolate ZS01 after 3 days of culture on V8-agar medium at 15, 20, 25, 30 and 35 °C.

**Figure 7 plants-12-01072-f007:**
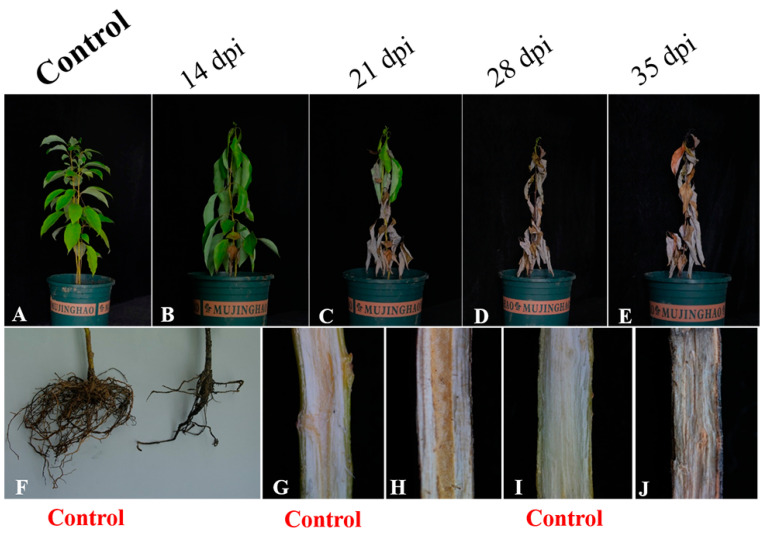
Symptoms on 2-year-old camphor after inoculation with isolate ZS01. (**A**) Control treated with sterile V8 liquid. (**B**–**E**) Diseased sapling, with symptoms at 14 days, 21 days, 28 days and 35 days post inoculation. (**F**) Control roots and roots of diseased saplings. (**G**,**H**) Control longitudinal section and longitudinal section from the stem of a diseased sapling 40 days after inoculation with ZS01. (**I**,**J**) Control longitudinal section and longitudinal section of the root of a diseased sapling 38 days after inoculation with ZS01. Dpi: days post inoculation.

**Figure 8 plants-12-01072-f008:**
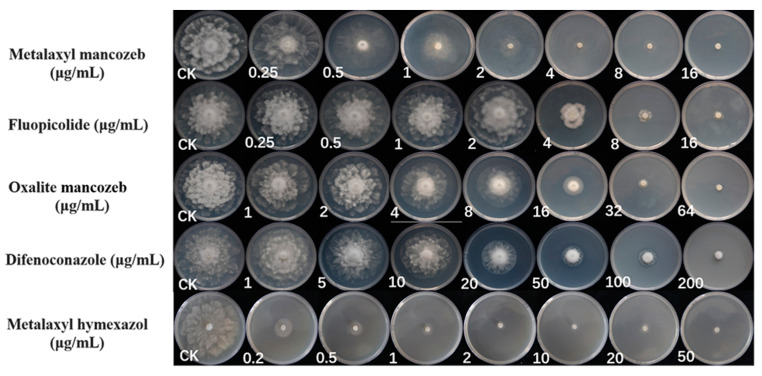
The effect of suppression on isolate ZS01 on plates by corresponding doses of five different fungicides (metalaxyl mancozeb, fluopicolide, oxalite mancozeb, difenoconazole, and metalaxyl hymexazol) based on fresh V8-agar medium for 3 days.

**Table 1 plants-12-01072-t001:** Mean half-maximal effective concentration (EC_50_ values) of isolates. (Data = mean ± standard error).

	Isolates		EC_50_ Values (μg/mL)		
	Metalaxyl Mancozeb	Metalaxyl Hymexazol	Fluopicolide	Difenoconazole	Oxalite Mancozeb
ZS01	1.31 ± 0.01	0.003 ± 0.002	2.55 ± 0.14	17.38 ± 3.56	8.32 ± 1.04
ZS02	1.29 ± 0.01	0.002 ± 0.002	2.35 ± 0.02	15.90 ± 2.88	9.40 ± 1.33
ZS03	1.30 ± 0.12	0.011 ± 0.004	2.62 ± 0.01	17.09 ± 3.24	10.23 ± 0.98
ZS04	1.28 ± 0.07	0.019 ± 0.001	2.50 ± 0.01	18.68 ± 3.66	7.88 ± 1.11

**Table 2 plants-12-01072-t002:** Primer sequences used for molecular identification of isolates.

Primer	Sequence (5′-3′)	PCR Conditions
ITS1 ITS4	TCCGTAGGTGAACCTGCGG TCCTCCGCTTATTGATATGC	Denaturation for 3 min at 94 °C, followed by 35 cycles; 30 s at 94 °C, 30 s at 55 °C, 60 s at 72 °C, and 10 min of a final extension at 70 °C
Coxlevup Coxlevlo	TCAWCWMGATGGCTTTTTTCAAC CYTCHGGRTGWCCRAAAAACCAAA	Denaturation for 3 min at 94 °C, followed by 35 cycles; 30 s at 94 °C, 30 s at 53 °C, 60 s at 72 °C, and 10 min of a final extension at 70 °C
CoxII-F CoxII-Rc4	GGCAAATGGGTTTTCAAGATCC TGATTWAYNCCACAAATTTCRCTACATTG	Denaturation for 3 min at 94 °C, followed by 35 cycles; 30 s at 94 °C, 30 s at 53 °C, 60 s at 72 °C, and 10 min of a final extension at 70 °C
NL1 NL4	GCATATCAATAAGCGGAGGAAAAG GGTCCGTGTTTCAAGACGG	Denaturation for 3 min at 94 °C, followed by 35 cycles; 30 s at 94 °C, 30 s at 55 °C, 60 s at 72 °C, and 10 min of a final extension at 70 °C
Btub-F1A Btub-R1A	GCCAAGTTCTGGGARGTSAT CCTGGTACTGCTGGTAYTCMGA	Denaturation for 5 min at 94 °C, followed by 35 cycles; 30 s at 94 °C, 30 s at 60 °C, 60 s at 72 °C, and 10 min of a final extension at 70 °C

**Table 3 plants-12-01072-t003:** GenBank accession numbers of isolates used for phylogenetic analysis in this study.

Species	Isolates	GenBank Accession Numbers				
		** *rDNA ITS* **	** *LSU rDNA* **	*coxI*	*coxII*	*β-tubulin*
*P. aichiense*	CBS 137195	AB948197	AB948194	AB948191	AB948192	AB948170
*P. chamaehyphon*	CBS 25930	AB690609	AB690593	AB690644	AB690674	AB948188
*P. carbonicum*	CBS 112544	AB725876	AB996605	AB690648	AB690678	AB948183
*P. citrinum*	CBS 119171	AY197328	AB690597	AB690649	AB690679	AB948180
*P. cucurbitacearum*	CBS 7496	AB725877	AB690598	AB690650	AB690680	AB948189
*P. delawarense*	382B	AB725875	AB690591	AB690642	AB690672	AB948181
*P. helicoides*	CBS 28631	AB725878	AB690594	AB690645	AB690675	AB948187
	H5	AB690611	AB690582	AB690633	AB690663	AB948186
*P. iriomotense*	GUCC0025	AB690622	AB690600	AB690652	AB690682	AB948172
	GUCC0028	AB690623	AB690601	AB690653	AB690683	AB948171
	GUCC0036	AB690624	AB690602	AB690654	AB690684	AB948173
	CBS 137104	AB690629	AB690607	AB690659	AB690689	AB948174
*P. litorale*	NBRC107451	AB690612	AB690583	AB690634	AB690664	AB948182
	CBS 118360	HQ643386	HQ665082	HQ708433	KJ595418	NA
*P. mercuriale*	CBS 122443	AB725882	AB690585	AB690636	AB690666	AB948179
*P. montanum*	CBS 111349	AB725883	AB690586	AB690637	AB690667	AB948184
*P. oedochilum*	CBS 25270	AB690618	AB690592	AB690643	AB690673	AB948175
	CBS 29237	AB690619	AB690595	AB690646	AB690676	NA
	GUCC5091	AB920534	NA	AB920493	AB920500	NA
*P. ostracodes*	CBS 76873	AY598663	AB690587	AB690638	AB690668	AB948178
*P. megacarpum*	CBS 112351	HQ643388	HQ665067	HQ708435	AB690665	NA
*P. boreale*	CBS 55188	HQ643372	HQ665261	HQ708419	AB690677	NA
*P. mirpurense*	CBS 124523	KJ831613	KJ831613	KJ831612	NA	NA
*P. vexans*	NBRC107442	AB690626	AB690604	AB690656	AB690686	NA
	NBRC107393	AB690630	AB690608	AB690660	AB690690	NA
	NBRC107380	AB690627	AB690605	AB690657	AB690687	NA
	NBRC107381	AB690628	AB690606	AB690658	AB690688	NA
	NBRC107397	AB690610	AB690581	AB690632	AB690662	NA
	CBS 11980	HQ643400	HQ665090	HQ708447	GU133518	NA
	P133	MT740895	MT729990	MT720669	MT720685	NA
	2D111	AB725880	AB856796	AB856784	AB948193	AB948169
	**ZS01**	OM663739	OP159402	OM791301	OM791302	OP185088
	**ZS02**	OM663740	OP159403	OM791303	OM791304	OP185089
	**ZS03**	OM663741	OP159404	OM791305	OM791306	OP185090
	**ZS04**	OM663742	OP159405	OM791307	OM791308	OP185091
	**ZS05**	OP941721	OP947167	OP948802	OP948803	OP948801
	**ZS06**	OP941722	OP947168	OP948805	OP948806	OP948804
	**ZS07**	OP941723	OP947169	OP948808	OP948809	OP948807
	**ZS08**	OP941724	OP947170	OP948811	OP948812	OP948810
	**ZS09**	OP941725	OP947171	OP948814	OP948815	OP948813
	**ZS10**	OP941726	OP947172	OP948817	OP948818	OP948816
	**ZS11**	OP941727	OP947173	OP948820	OP948821	OP948819
	**ZS12**	OP941728	OP947174	OP948823	OP948824	OP948822
	**ZS13**	OP941729	OP947175	OP948826	OP948827	OP948825
	**ZS14**	OP941730	OP947176	OP948829	OP948830	OP948828
	**ZS15**	OP941731	OP947177	OP948832	OP948833	OP948831
	**ZS16**	OP941732	OP947178	OP948835	OP948836	OP948834
	**ZS17**	OP941733	OP947179	OP948838	OP948839	OP948837
	**ZS18**	OP941734	OP947180	OP948841	OP948842	OP948840
	**ZS19**	OP941735	OP947181	OP948844	OP948845	OP948843
	**ZS20**	OP941736	OP947182	OP948847	OP948848	OP948846
	**ZS21**	OP941737	OP947183	OP948850	OP948851	OP948849
	**ZS22**	OP941738	OP947184	OP948853	OP948854	OP948852
	**ZS23**	OP941739	OP947185	OP948856	OP948857	OP948855
	**ZS24**	OP941740	OP947186	OP948859	OP948860	OP948858
	**ZS25**	OP941741	OP947187	OP948862	OP948863	OP948861
	**ZS26**	OP941742	OP947188	OP948865	OP948866	OP948864
	**ZS27**	OP941743	OP947189	OP948868	OP948869	OP948867
	**ZS28**	OP941744	OP947190	OP948871	OP948872	OP948870
	**ZS29**	OP941745	OP947191	OP948874	OP948875	OP948873
	**ZS30**	OP941746	OP947192	OP948877	OP948878	OP948876
*Phytophthora nicotianae*	GF101	AB688384	AB688538	AB688265	NA	AB948190

NA = Information is not available. Species and isolates obtained in this study are shown in bold.

## Data Availability

All data generated or analyzed during this study are included in this article.

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
