# Peer review of "Root Rot of *Cinnamomum camphora* (Linn) Presl Caused by *Phytopythium vexans* in China"

_plants, 2023, doi:10.3390/plants12051072_

Round 1

Reviewer 1 Report

Manuscript expands knowledge on Phytopythium vexans especially in host range of these pathogen. Performed experiments conclusively proves that the species is the cause of root rot of Cinnamomum camphora. The authors conducted all the necessary tests in accordance with the principles of microbiological art, enabling the isolation, identification and confirmation of the causative role of the obtained isolate in causing the disease.

Author Response

Dear Reviewer:

Thank you for taking the valuable time to conduct a professional evaluation of our manuscript. Your comments will help us to further improve the manuscript. We would like to express our sincere thanks to you.

Sincerely yours

Feng-Mao Chen Ph.D.

Professor

Nanjing Forestry University

159 Longpan Road

Nanjing, Jiangsu 210037

China

E-mail: cfengmao@njfu.edu.cn

Reviewer 2 Report

Dear Editor, Authors,

The work presented in the reviewed manuscript is interesting and has the potential to contribute to the current knowledge, bringing interesting results pertaining a root rot disease of economically valuable tree species in Asia. For this reason, it deserves to be published in journal Plants.

General comment: I suggest to include in the first chapter in the Results detail about field investigation regarding presence of symptoms, disease severity and incidence, infection rate, mortality to get information about harmfulness of the pathogen to studied host tree, as this is a first detection of pathogen on camphor and P. vexans is worldwide distributed pathogen there is high potential for further infection of camphor trees in other areas.

l- 16: abbreviate the genus name, P. vexans

Introduction

l. 31: after province I suggest to include country: China

l. 63: Phytopythium.- delete dots

l. 64, 65 use Latin names for plant species

Material and Methods

Chapter 2.1 include in the chapter:

1.information or details about camphor plot (size of area, age, number of trees etc.), 2.information about management (as in the Results there are mentioned consequences regarding this), 3.observation of symptoms,

l. 106-107: Latin names change in Italic

l. 109 corrected[30]- missing a space

l. 123-125 The pathogen had... remove, delete this information from M&M, this is result

l- 139-140 try to combine in one sentence, to eliminate the information about five temperatures in one

l. 144- rephrase: Four isolates selected for morphological and biological identification were used in pathogenicity tests

l. 155 “The experiments were conducted three times.”- simultaneously? Or in some time span? In l. 158 the same information, enough to say once?

l. 158: which primers were used/genes were amplified? Not important to describe the procedure, but you should to refer to chapter 2.2 as described above

Chapter 2.5- from the text is not clear why or how the final concentrations of fungicides were selected

Results

l. 185-186: this your result?

l.186-187: Increase of dead trees each year – specify to see the seriousness of the disease and/or the pathogen; each year- field survey was done only in 2021 according to M&M; some camphor- specify have many, the part with “good” management stay healthy?

l. 196-197: One oomycete... this not a field survey result, rename the chapter, start a new paragraph where you put results about obtained cultures, isolation or move to new chapter, no information about cultures were provided

chapter 3.2 Phytopythium, P. vexans – italic, check Latin names in this chapter and change font to Italic

l. 263: delete this part of sentence, this was said in M&M: According to observations and measurements under 263 a 40x microscope,

l. 272: 25-30C – you mean range – temperature between 25-30C or exactly the two temperatures, at 25C and at 30C

l. 281-283: rephrase to remove methodological part

l. 292-294: Fig 7G is inoculated, 7H healthy in the text, similar comment for 7I and 7G, in the Fig as control is marked G and I, correct it

l. 302: these symptoms – change to: The symptoms on artificially inoculated seedlings were...

Fig 7: what means dpi above the pictures? Explain in the figure legend

l. 311: have you evaluate the difference between isolates statically,

Discussion

l.325: delete the first sentence, this was already mentioned in Introduction

l. 327-328: what you mean by this, is it your result? I did not register this information in Results,

l. 329, l. 331 Latin name Italic

l.344: industrial crops? Ornamentals?

l. 355-356: this part of paragraph about temperature had no reference, please add it, it can’t be your own result as your experiment was conducted in one stand

l. 375- Phytopythium. Vexans- delete dot and not “disease” but soil borne pathogen

l.386: give more details, discuss closer your results and the previously published

l.388-389: also, in planta experiment, that is missing in the current study

Author Response

Dear Reviewer:

Thank you for your decision and constructive comments on my manuscript. We have carefully considered the suggestion of Reviewer and make some changes. We have tried our best to improve and made some changes in the manuscript, the detailed corrections are listed in attachment.
